# Fog Classification by Their Droplet Size Distributions: Application to the Characterization of Cerema's Platform

**Pierre Duthon** *,† , **Michèle Colomb** † and **Frédéric Bernardin** †

Cerema, Equipe-projet STI, 8-10, rue, Bernard Palissy, CEDEX 2, F-63017 Clermont-Ferrand, France;
Michele.Colomb@cerema.fr (M.C.); frederic.bernardin@cerema.fr (F.B.)
* Correspondence: pierre.duthon@cerema.fr; Tel.: +33-4-7342-1069
† These authors contributed equally to this work.

**Abstract:** Fog is one of major challenges for transportation systems. The automation of the latter is based on perception sensors that can be disrupted by atmospheric conditions. As fog conditions are random and non-reproducible in nature, Cerema has designed a platform to generate fog and rain on demand. Two types of artificial fog with different droplet size distributions are generated: they correspond to radiation fogs with small and medium droplets. This study presents an original method for classifying these different types of fog in a descriptive and quantitative way. It uses a new fog classification coefficient based on a principal component analysis, which measures the ability of a pair of droplet size distribution descriptors to differentiate between the two different types of fog. This method is applied to a database containing more than 12,000 droplet size distributions collected within the platform. It makes it possible to show: (1) that the two types of fog proposed by Cerema have significantly different droplet size distributions, for meteorological visibility values from 10 m to 1000 m; (2) that the proposed droplet size distribution range is included in the natural droplet size distribution range; (3) that the proposed droplet size distribution range should be extended in particular with larger droplets. Finally, the proposed method makes it possible to compare the different fog droplet size distribution descriptors proposed in the literature.

**Keywords:** fog; meteorological optical range; droplet size distribution

## 1. Introduction

Fog is one of the major challenges for transportation systems. The economic impact is comparable to that of tornadoes [1]. For land transport systems, the first impact concerns road safety. In the presence of fog the Meteorological Optical Range (MOR) is reduced. This leads to an increase in the number of accidents at night and doubles the number of fatalities per 100 accidents [2]. Fog conditions also lead to a reduction in speed, and thus have an impact on mobility. Regarding the study of human perception through fog, only the macroscopic density, defined by the MOR, denoted as $V$, is to be taken into account [3]. Fog is considered to be present for a MOR below 1000 m [3]. The MOR is the distance through fog for which the luminous flux of a collimated light beam is reduced to 5% of its original value.

With the advent of Advanced Driver Assistance Systems (ADAS), and the development of autonomous vehicles, it is very important to take into account the impact of fog on the sensors dedicated to driving. The MOR is always critical, however, as sensors can use different wavelengths, some of which are beyond the visible; the microstructure of the fog is also critical [4]. There are indeed many different Droplet Size Distributions (DSD) of fog [1]. However, fogs with different DSDs do not

have the same formation mechanisms and it is therefore difficult to find them in the same place and at the same time. Moreover, although significant measurement campaigns have been carried out [5–7], the fog phenomenon is very random and the densest fogs are quite rare [8]. It is therefore difficult to test the sensors on board vehicles in the natural environment, with real fogs, all the more so as the meteorological sensors must then be located close to the roads.

These different factors led Cerema to propose a test platform that can be used to reproduce fog conditions on demand. Cerema is a state agency of 3000 employees, placed under the supervision of the ministry for Ecology, Energy and Sustainable Development and Energy and the ministry for Regional Equality and Housing. The Cerema acts as a resource center for scientific and technical expertise. Specific features of Cerema are a strong territorial rooting and its capacity to combine many fields of expertise to answer complex questions related to sustainable development. In the field of mobility and transportation, the Research Team, Intelligent Transport System, (Equipe-projet STI) at the Laboratory of Clermont-Ferrand (one of the 17 laboratories of Cerema) conducts research on the mobility and safety in adverse weather conditions as fog and rain. The study of the impact of reduced visibility conditions on driving is addressed both from a technical approach and from driver perception. Tests and analyses are carried out in the platform. That is a research infrastructure, unique in Europe, open to research institutes as well as to private companies in order to evaluate performances of smart system dedicated to transport, or to proceed to any other scientific activities such as model validation, perception tests.

Cerema's Adverse Weather platform [9] has been allowing research teams to come and test their system since 1984. A number of studies have been published [10–17]. The Adverse Weather platform can reproduce various fogs with two types of DSD and for MORs from 10 to 1000 m. Although initial work has shown a coherence between natural fogs and fogs reproduced in the platform [7,18], it is now necessary to validate the DSD of the platform fogs on a larger scale. This is the subject of this article.

Concerning fog DSD analysis, the literature is very extensive. The reader will be able to consult more complete states of the art on this subject [1,19]. All the studies show that fog droplet size ranges from a few tenths of a micron to a few tens of microns [1,20,21]. Many of the previous studies use coefficients calculated on the DSD to characterize it. Examples include the liquid water content (LWC) [6,7,9,22–35], the total concentration of drops, $N_{tot}$ [6,7,9,22–31,33–36] which are systematically calculated, the mean diameter $D_{mean}$ [7,9,22,24–27,29,31,33–35,37] or its variants [7,9,22,26,28,29,31,36]. Finally, original parameters such as different order moments on the distribution such as standard deviation, skewness and kurtosis, or the autoconversion rate are sometimes used [33–35]. Some of these studies attempt to establish a relationship between the above-mentioned parameters: $N_{tot}$ vs. LWC [38,39]. Other studies attempt to characterize the DSD by modeling them. Two main categories of laws are used for fitting: shifted gamma laws [40–43] and log normal laws [7,18,41].

In spite of this very rich bibliography, the classification of fogs is still very complicated. Some works qualitatively discuss the value of coefficients combinations in relation to each other [33–35]. Howerver, to our knowledge, there is no quantitative analysis of the relevance of some coefficients to others, nor any systematic method for classifying fogs according to their DSD. This may also be due to small data sets. It is for these reasons that this study is proposed. The main goals are:

- to characterize the two types of fogs of the platform. The objective is to show that these fogs are significantly different, over a large amount of data.
- to show that these two types of fog DSDs are similar to natural fogs.

## 2. Cerema's Platform

### 2.1. Presentation of the Platform

The Cerema Adverse Weather platform was developed to investigate all transportation systems that could be affected by adverse conditions, including fog and rain [4,9,44]. This 31 m long platform includes a 15 m fixed section (tunnel) and a 16 m greenhouse with a transparent envelope for daytime

conditions and an opaque cover for night time conditions. This platform is 5.5 m wide and 2.3 m high. An observation post is located at the end of the platform, in order to install the equipment outside of the wet area during testing. It can be used to reproduce various scenarios, such as detection of vulnerable road users or fixed obstacles, in day or night conditions, with many targets and with various ranges of fog and rain precipitations over a total length of 30 m (Figure 1) [45].

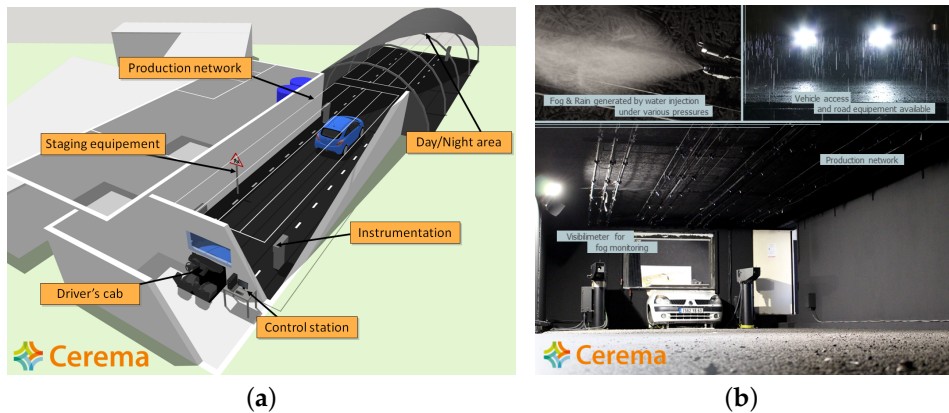

|            (a)            |            (b)            |

**Figure 1.** Cerema's Adverse Weather platform [46]. (**a**) Overall structure. (**b**) Instruments and nozzles.

The platform is dedicated to research and development and is also available to private companies looking for a testing facility with controlled conditions. It has been used for years in partnerships and on collaborative projects in order to investigate various scientific topics, such as human perception in adverse conditions [10,11], vision system capabilities in fog or rain conditions [12–14] or computer vision algorithms for object detection [15–17].

This platform has a high-level of instrumentation to evaluate the performance of perceptual sensors in adverse conditions. Some weather instruments are dedicated to characterizing the atmosphere in fog or rain conditions (Table 1). In order to characterize the road environment itself and the object properties to be perceived by humans or by sensors, other devices are also available (Table 1).

**Table 1.** Cerema's Adverse Weather platform intruments.

| Device | Model | Measure |
|---|---|---|
| Transmissometer | Degreane Horizon TR30 | MORs from 10 to 500 m $\pm$ 1% and from 500 to 1000 m $\pm$ 5%. |
| Particle Size Analyser | PALAS WELAS 2100 | DSD of fog over the range 0.3–17 μm in 60 classes of diameter in this range. 10% uncertainty for total concentration above $10^5$ cm$^{-3}$. |
| Rain gauge | LSI DQA136 | Rainfall rate from 0.2 to 60 mm h$^{-1}$ $\pm$ 0.2 mm h$^{-1}$ and from 60 to 600 mm h$^{-1}$ $\pm$ 1%. |
| Spectro-pluviometer | OTT Parsivel | Rainfall rate from 0.001 to 1200 mm h$^{-1}$ $\pm$ 5%, rain DSDs and velocity. |
| Video-photocolorimeter | TECHNOTEAM LMK 98-4 | Luminance from 0.003 to 50,000 cd m$^{-2}$ in the visible range. |
| Reference cameras | Xenics | Visible, near-infrared, short-waves infrared and long-wave infrared bandwidths. |
| Spectroradiometer | Spectral Evolution PSR3500+ | Spectral radiance in the range 350 to 2450 nm. |

*2.2. An Example of Platform Use Cases*

Cerema's Adverse Weather platform has recently been used to set up a model to predict under what fog level a Time of Flight Light Detection And Ranging (ToF LiDAR) is capable of detecting targets [47].

By transmitting lasers and processing laser returns, LiDAR perceives the surrounding environment through distance measurements. Because of high ranging accuracy, LiDAR is one of the most critical sensors in autonomous driving systems. Using LiDAR data, a lot of algorithms have been developed for object detection/tracking, environmental mapping, or localization. However, a LiDAR's ranging performance suffers under adverse weather (e.g., fog, rain, snow etc.), which impedes full autonomous driving in all weather conditions. For analyzing the performance of a typical LiDAR (Velodyne UltraPuck) under a fog environment, the LiDAR was installed on the platform, and typical road objects were placed in front of it. The objects used do not have the same LiDAR reflectivity (Figure 2a), and they were successively moved onto the platform. For each position, dense fog was generated and dissipated to obtain measurements for MOR ranging from 10 to 400 m. As shown in Figure 2b, there is a lower MOR threshold at which the LiDAR no longer detects objects. This threshold depends on the reflection of the object and the distance between the object and the LiDAR. This threshold is called disappearing visibility. All the tests carried out have thus made it possible to train a machine learning-based model for predicting the disappear visibility as a function of the nature of the object (LiDAR reflectivity) and its distance from the LiDAR. Thus, the systems installed on board vehicles can now, depending on fog conditions, know their ability to detect objects and thus adapt the vehicle's behavior according to the conditions. The experimental results and methods deployed in Cerema's Adverse Weather platform are therefore helpful for ToF LiDAR specifications from automotive industry.

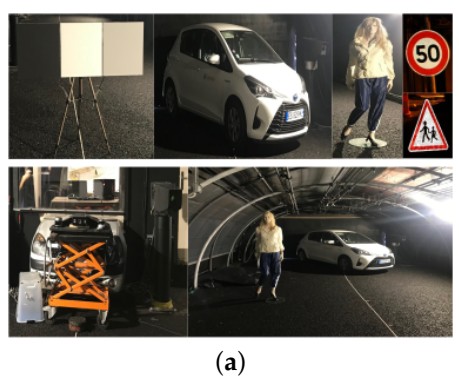
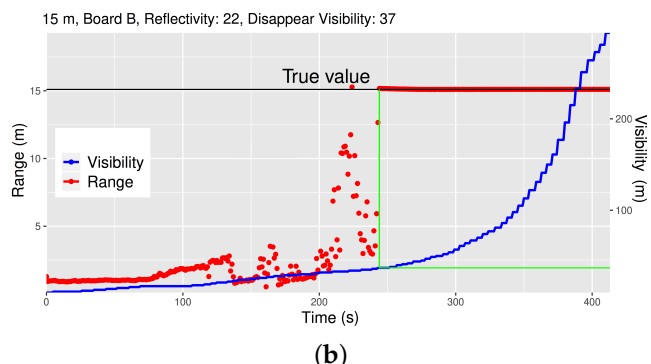

(**a**) (**b**)

**Figure 2.** Use case of the Cerema's Adverse Weather platform for the qualification of a Time of Flight Light Detection And Ranging (ToF LiDAR) [47]. (**a**) Top: used targets (three calibrated boards, vehicle, a dummy model and two traffic signs). Bottom: Velodyne UltraPuck on the table and an example scenario. (**b**) Average range measures (LiDAR detection) with Meteorological Optical Range (MOR) $V(t)$ for randomly selected individual lasers of the board B target, at 15 m distance. Disappear visibility is marked as the crossings of green line.

### 2.3. Mechanical fog Production System and Limitation of This Study

At the Cerema Adverse Weather platform, a mechanical process allows the production of fog. The production system consists of of a high-pressure booster giving a range between 30 and 100 bars. This booster is connected to water pipes by means of magnet valves allowing short pulses of fog. A double water tank controls the quality of water, thanks to water filtering and de-mineralizing systems. Different DSDs can be produced by changing the nature of the water injected (normal tap water or demineralized water). The nozzles are fixed to water pipes by means of extenders that enable the water flow to be oriented for a better homogeneity of fog production in the platform. Two separate circuits enable pin nozzles or vortex nozzles to be used alternatively (Figure 1b). The water is forced under high pressure into the spray nozzles, which atomizes the droplets and produces a thin water spray. The flow delivered by the two types of nozzles according to working pressure, is in the range of 2 to 6 L h$^{-1}$. As the production process is mechanical and not thermodynamic, it is independent of the temperature. The working conditions are between 5 °C and 25 °C. Temperature may influence the stability of fog when produced, and the fog density. This is balanced by real-time adjustment by remote control

of the fog production during the test, in order to stabilize the MOR. It is therefore possible to produce fog of different densities by modifying the quantity of water injected, in all temperature and humidity conditions. MOR (visibility) is measured by the transmissometer and can be kept constant by 10 m increments between 10 m and 100 m.

The mechanical process for producing artificial fog generates very small humid particles, that could be considered as wet aerosol (less than 2 µm), in great concentrations at the same time as larger fog droplets (over 2 µm). The high total concentration is mainly due to the wet aerosol as previously described [9]. These characteristics have been reported in some natural continental fogs, having a large number of submicron wet aerosol [36,48]. It was also found that aerosol–fog continuity in polluted areas [49]. Furthermore, it appears that during the transition phase between aerosol and fog droplets, swollen aerosols are reducing the visibility [50].

The wet aerosol component of natural fog can be observed only if the sensor used has a measuring range less than 2 µm in particle diameter. Cerema's Adverse Weather platform, with it mechanical process, is thefore considered to produce mostly continental fog.

The setting of the production system, of the Adverse Weather platform, allows us to produce two different continental fog DSD, called small droplets fog (SD fog) and medium droplets fog (MD fog), in the investigated range of droplet size (0.5 to 17 µm) with PALAS WELAS 2100 sensor. Figure 3 shows an example of DSD with natural fog, both types of fog produced within the platform, and the most common models in the literature [40,41]. On the Figure 3, Cerema Small Droplets and Cerema Medium Droplets, correspond to the two fogs produced in the platform. The Paris Fog graphs correspond to natural data acquired during the Paris Fog measurement campaign [5–7]. During this campaign fogs with a single wet aerosol/small droplets mode (diameter around 1 µm) and fogs with two modes with small and medium droplets (diameter around 1 µm and 5 µm) could be recorded. The models of literature are Shettle's models, which are composed of advection and radiation fog (Shettle Advection 1, Shettle Radiation 4), and Deirmendjian haze-like fogs (Deirmendjian Haze M and Deirmendjian Haze L). The addition of the latter is important because the platform fogs have small droplets, similar to wet aerosol, compared to the literature about fog. The DSDs proposed in the Figure 3 are comparable, as they all correspond to an MOR in the range 260–270 m. According to the Figure 3, there are even major differences between the different models in the literature. Shettle's models always tend to overestimate drop size while Deirmendjian's models tend to underestimate drop size. Natural and platform fogs are quite close.

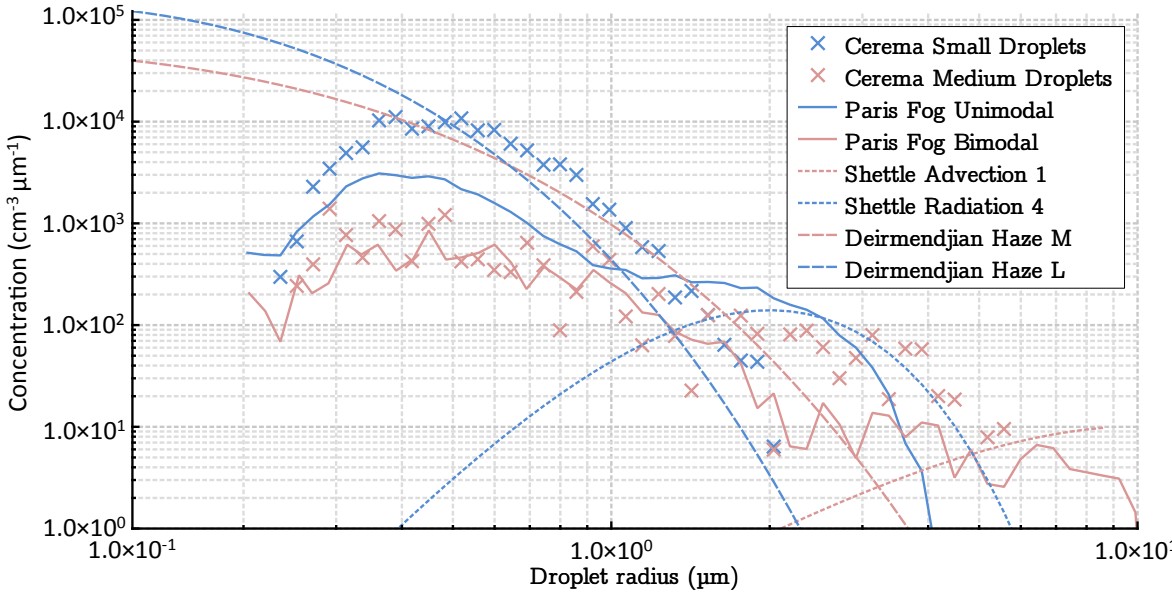

**Figure 3.** Examples of droplet size distribution.

It should be noted before going any further that the Particle Size Analyser (PSA) we are using in this study (PALAS WELAS 2100) has a range of 0.3–17 μm. This range is not sufficient to address all types of fogs [6,51,52]. We show in Figure 4 the extinction coefficient at 550 nm calculated by Mie theory with 4 models of natural fog DSDs mentioned in Figure 3 (Shettle Advection 1, Shettle Radiation 4, Deirmendjian Haze M and Deirmendjian Haze L). More precisely we consider a truncation diameter *D* represented on the x-axis for each DSD and then we calculate the corresponding extinction ratio as the extinction for a DSD truncated at *D* μm over the extinction for a DSD truncated at 50 μm. The truncation diameter *D* therefore makes it possible to assess the impact of the PSA measurement range on the extinction estimation. The upper limit value of 50 μm is chosen as the limit of classical PSAs like Fog Monitor or CDP [6,51,52]. We can observe in Figure 4 that a DSD obtained by a truncation diameter of 17 μm (corresponding to the PALAS WELAS 2100 sensor) explained the whole extinction except for the Shettle advection model (less than 10 % of the extinction is explained by the PALAS WELAS 2100 sensor in this case), which is not in the scope of radiation continental fogs produced in the platform and addressed in this paper. These figures are consistent with findings in the literature [6,51,52]. This observation remains true, even taking into account the fact that the PALAS WELAS 2100 sensor would have difficulty detecting drops above 8 μm [6,51]. Even in that case, more than 82% of the extinction is still explained for all radiation models (see Figure 4), keeping apart Shettle's advection model. Our study is therefore limited to radiation fogs or haze-like fogs. Advection fogs will be investigated later on by using other sensors having a measuring range extended to larger droplets up to 40 μm.

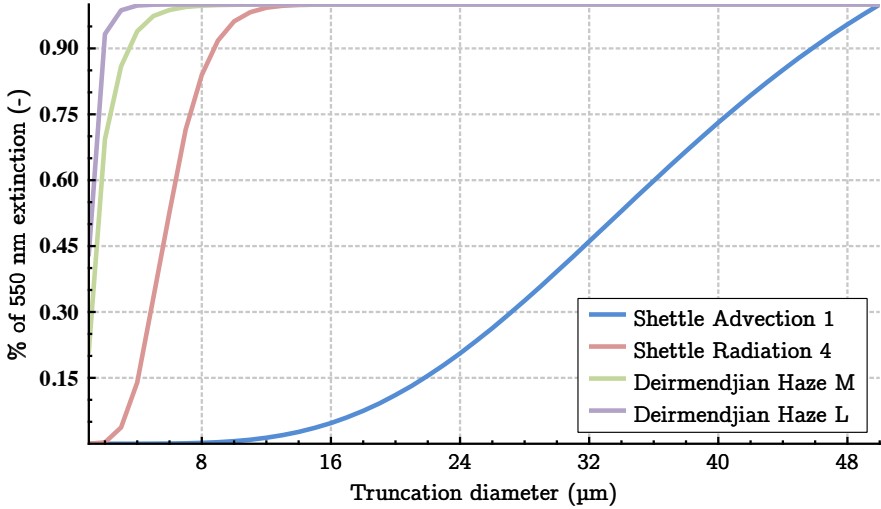

**Figure 4.** Percentage of 550 nm extinction versus truncation diameter (*D*).

Aside from the notable differences that can be observed between the different DSDs in Figure 3, there is to our knowledge no method to quantify the difference between two DSDs. It is not acceptable to rely on a single example of DSD (given in Figure 3) to compare our platform's artificial fogs to natural fogs, which are much more varied than those of Paris Fog. Therefore, at the moment, there no method to show that the two fogs reproduced within the platform are: (a) different from each other; (b) similar to natural radiation fogs in the literature. The purpose of this paper is to propose a method for answering these questions. It will first focus on a descriptive and qualitative analysis based on the use of DSD descriptors, and then on a quantitative analysis with the use of a new metric to assess the quality of differentiation of the DSD descriptors examined. This work will be carried out on the common descriptors in the literature (LWC, mean diameter, $N_{tot}$, etc.).

## 3. Methods

### 3.1. Choice of Parameters

A DSD is represented by a distribution $n(r)$ which describes the number of particles (expressed as a number per $cm^3$) of radius $r$ divided by the class width $dr$. $n(r)$ is thus usually expressed in $cm^{-3}$ $\mu m^{-1}$. Expressing the concentration in this way makes it possible to overcome the class width, as the class width is often not constant in PSA's data. Since PSAs have limits, the DSD is truncated between $r_{min}$ and $r_{max}$.

As detailed in the introduction, much work in the literature proposes to calculate parameters to characterize fog DSDs. Among them, the most frequently encountered are the following [8].

Given a fog with a DSD $n(r)$, the extinction coefficient $\beta$ (Equation (1)) measures how much the fog attenuates light radiation from the visible range. It is used to relate to the MOR, noted as $V$, as shown in Equation (3). It is based on Mie's theory, which gives an extinction coefficient $Q_{ext}(r)$ for each drop of radius $r$ [4]. $\beta$ is usually expressed in $m^{-1}$. It can be related, for wavelengths in the visible range, to $LWC$ and $D_{eff}$ as proposed in Equation (2) [53].

$$\beta = \int_{r_{min}}^{r_{max}} \pi Q_{ext}(r)n(r)r^2 dr \tag{1}$$

$$\beta = \frac{3LWC}{\rho_w D_{eff}} \tag{2}$$

where $\rho_w$ is the density of liquid water.

$$V = -\frac{ln(0.05)}{\beta} \simeq \frac{3}{\beta} \tag{3}$$

where $ln(0.05)$ corresponds to a contrast of 5%.

The total number of drops $N_{tot}$ involves summing the number of droplets in all drop size classes of a given particle size. $N_{tot}$, defined in Equation (4), is usually expressed in $cm^{-3}$.

$$N_{tot} = \int_{r_{min}}^{r_{max}} n(r)dr \tag{4}$$

The Liquid Water Content $LWC$ corresponds to the concentration of water in the air. It is generally expressed in $g\,m^{-3}$ from the third moment of the DSD (Equation (5)).

$$LWC = \int_{r_{min}}^{r_{max}} \frac{4\pi\rho_w}{3} n(r)r^3 dr \tag{5}$$

The total surface area, expressed in $m^2\,m^{-3}$, comes from the second moment:

$$S = \int_{r_{min}}^{r_{max}} 4\pi n(r)r^2 dr \tag{6}$$

The mean diameter $D_{mean}$ is the first-order moment of the DSD. It is defined by the following equation:

$$D_{mean} = 2\frac{\int_{r_{min}}^{r_{max}} n(r)r dr}{N_{tot}} \tag{7}$$

Similarly, $D_{quad}$ (resp. $D_{vol}$) is the second-order (resp. third-order) moment:

$$D_{quad} = 2\frac{\int_{r_{min}}^{r_{max}} n(r)r^2 dr}{N_{tot}} \tag{8}$$

$$D_{vol} = 2\frac{\int_{r_{min}}^{r_{max}} n(r)r^3 dr}{N_{tot}} \tag{9}$$

Finally, the effective diameter (or equivalent) relates *LWC* to *S*. It corresponds in a way to the average apparent surface area of the water droplets and is defined by the following equation:

$$D_{eff} = \frac{D_{vol}}{D_{quad}} = \frac{6LWC}{\rho_w S} \tag{10}$$

*3.2. Methodology*

Our main objective is to quantitatively measure the difference between the two kinds of fog produced within the Cerema Adverse Weather platform using a statistical approach. Huge amounts of data are required for this. Our aim is also to show how the fog produced in the platform is repeatable. We, therefore, chose to use data from different test sessions. In the European DENSE project [54], numerous tests have been carried out within the platform [4,8,13,16,17,55–57]. For this study, we have therefore chosen to use all the meteorological data acquired during these trials.

The measurement of fog characteristics is set up in Cerema's Adverse Weather platform. It uses the PALAS WELAS 2100 sensor and the Degreane Horizon TR30 transmissometer, whose complete characteristics are given in the Table 1.

Throughout the tests, the sensors are all placed in the same horizontal plane, 1.20 m above the ground. The timestamps of all sensors are synchronized.

During tests in fog conditions, two types of water can be used. Previous studies have shown that changing the type of water has a qualitative effect on the particle size [9]. The two fogs produced are therefore:

- small droplets fog (SD fog), with a main diameter mode around 0.5–1 µm, produced with normal tap water.
- medium droplets fog (MD fog), with two diameter modes around 0.5–1 µm and 7–10 µm, produced with demineralized water.

Although internal analyses revealed a difference between these two types of fog, and qualitative differences were shown [9], no quantitative results had been published. The proposed protocol should therefore be able to provide evidence that the two patches of fog produced within the platform have different DSDs, based on all the data collected during the various previous tests.

Two devices measuring three different quantities are used during the tests carried out. In order to perform the analysis, the data of the three sensors must be linked by their time stamp. A database containing the following information is created:

- fog type (SD or MD),
- MOR (temporal resolution of one second),
- DSD (temporal resolution of one second).

Measuring DSD is difficult because huge numbers of droplets are counted in a very small volume. At one-second timesteps (temporal resolution), the DSD data was thus too variable. So these data must be smoothed out. Internal tests have shown that a 60 s smoothing is relevant for standard applications. However, we will here use a 10 s smoothing, in order to retain high variability but also to be relevant in high MOR classes (between 500 and 1000 m) for which the fog varies more quickly. 12,004 DSDs were obtained within the platform. They were all used in this article. These data were recorded from December 2017 to April 2019 during five completely independent measurement campaigns as part of the DENSE project [4,8,13,16,17,55–57]. This will show how the fog produced within the platform is repeatable.

Before using the collected data, two filters were applied. The aim is to keep the data of the best quality, in order to obtain the most accurate results possible. The first filtering applied consists

in checking the correspondence between the MOR measured by the TR30 (which serves as a reference), $V_{ref}$ and the MOR estimated from the DSD, $V_{DSD}$. Using Mie's theory, it is possible to calculate the MOR for a given DSD [4]. In this study, only the data respecting the Equation (11) are kept. So 5128 DSDs, for which reliability is assured, are retained.

$$0.5 \leq \frac{V_{ref}}{V_{DSD}} \leq 2 \tag{11}$$

A second filtering is performed to ensure that the safest data are kept. This is based on the use of the standard deviation. This allows for the filtering of abnormal data assuming a Gaussian law is followed. The method is as follows: the standard deviation is calculated for each proposed descriptor. The data are filtered by retaining only those for which all the values of the descriptors taken into account are within plus or minus three times the standard deviation. This makes it possible to filter out outliers, at the most extreme values, while retaining 99.7% of the data. This filtering, therefore, depends on the set of descriptors analyzed. After the latter, 4970 DSDs acquired from the platform were retained for further analysis.

Most of the data are therefore those collected from the two artificial patches of fog of the platform. Data from natural fogs were also used for comparison. These data come from the Paris Fog measurement campaign [5–7]. They are measured with the same measuring instrument on SIRTA site, a PALAS WELAS 2100. They have one- and two-mode DSDs. The MOR of the fogs concerned vary between 50 m and 14,000 m with a large percentage of them ranging between 100 m and 1000 m. More recent studies [52] have been carried out on the SIRTA site as part of the continuation of the Paris Fog project. These studies use PSAs complementary to PALAS WELAS 2100 (such as the FM-100), which allows the measurement of drops up to 50 μm in diameter. However, we only use PALAS WELAS 2100 data from the first campaign [6], in order to obtain data comparable to those we measure in the platform, with the same sensor.

As explained in the introduction, many DSDs descriptors are proposed in the literature. Some DSD classifications relate to physical parameters: they will be referred to as conventional parameters in the following. These parameters include, for example, the liquid water content (LWC), the total concentration of drops, $N_{tot}$ or the mean diameter $D_{mean}$. It is relevant to compare these descriptors as they are often used in the literature, but not consistently. It should be noted that some of them are more or less correlated [38,39].

In this study, we chose to work on parameters calculated from the DSDs (conventional parameters) rather than on the DSDs themselves. It could have been envisaged to analyse the DSDs directly, taking into account each class as an input parameter. This choice was not made because the results obtained would have been too dependent on the PSA, in particular on its measuring range (minimum and maximum class value) and resolution (class width). It is therefore proposed to use parameters calculated on the DSDs rather than using the DSDs directly. This makes it possible to be independent of the PSA used, and thus makes the proposed method expandable. It should be noted, however, that in order to generalize the proposed method, care should be taken to truncate the DSDs over the same diameter measurement range. This is because some PSAs are able to count droplets in more or less wide ranges, which induces a shift of mean values ($D_{mean}, \ldots$) or global counts ($N_{tot}, \ldots$) [6,7,9,51].

In this paper, the following approach was applied:

- A study of the correlation between the parameters was made.
- This resulted in a restriction of the parameters analyzed. It is important to limit redundancy in the parameters used as much as possible.
- Once the parameters are limited in number, a qualitative and descriptive analysis was proposed. It is based on a graphical comparison by pairs of parameters. Whenever possible, data from the literature were added to the data from the platform. Data from natural fogs were also added in order to measure the correspondence between the fogs produced within the platform and natural fogs.

- In order to be able to compare the parameters against each other, a principal component analysis (PCA) will be performed. This will provide the most relevant combinations of parameters to sort the two types of fogs produced in the platform. This will also make it possible to verify how a type of parameter classifies the two types of fogs produced within the platform.
- The two descriptive and qualitative analyses are followed by a quantitative analysis based on a metric. The latter consists of calculating a coefficient, known hereafter as a Fog Classification Coefficient (FCC). This coefficient is calculated along the lines of the first round of the K-mean method. The K-mean method is an unsupervised classification method. It involves classifying a set of data into *N* groups, without labelled data for learning. The FCC is calculated in the following way for a given pair of parameters (Figure 5) :

  - The input data for the calculation of the FCC are the coordinates according to two chosen parameters, calculated on each of the DSDs in the database. In Figure 5a, the two types of fogs (crosses and circles) are drawn along two axes corresponding to two parameters and each point corresponds to one DSD in the database.
  - The first step is to average the data from each of the two types of fog (Figure 5b). This average is taken as the centre of each group. In Figure 5b, the black cross is the average of crosses and the black circle is the average of circles.
  - The second step consists of calculating the distance to the centre of each group for each data point. The two distances obtained (distances to the centre of the group of MD and SD fogs) are compared for each data point. The data point is then assigned to the group with the smaller distance (Figure 5c). In Figure 5c, each group is on either side of the black line.
  - Finally, once all data points are assigned to one of the two groups, the number of well-classified points is calculated (Figure 5d). The FCC is the rate of well-classified data points. In Figure 5d, well-classified points are green and wrongly classified points are red. It should be noted that for the calculation of the FCC, here we use a supervised method (with known label), although the original K-mean is unsupervised.

- One FCC is then obtained per pair of parameters, or by pairs of PCA vectors. The FCCs vary between 0 and 1, and the closer the value is to 1, the more effectively the pair of parameters taken into account allows for efficient separation of the two fog types (MD and SD). This classification coefficient has many advantages. This metric is in fact not dependent on units, nor on orders of magnitude (one parameter can vary between $-10$ and 10 while another can vary between 0 and $10^9$). It, therefore, allows an impartial and quantified comparison of the very varied and heterogeneous fog DSD parameters proposed in this article.

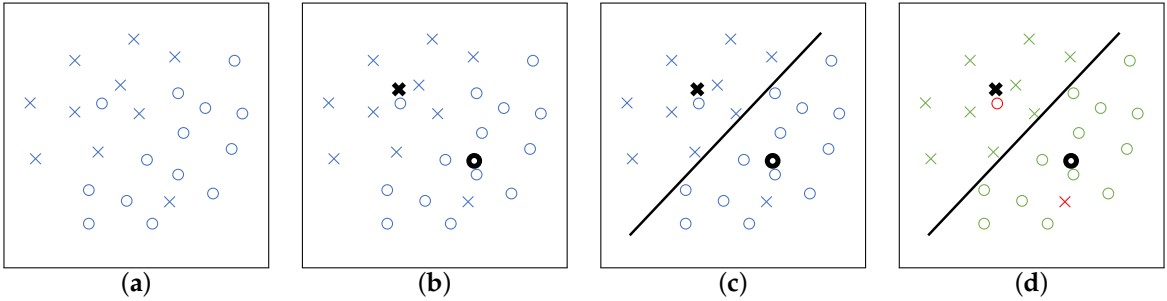

**Figure 5.** Principle of calculation of the Fog Classification Coefficient (FCC). (**a**) Two type of fogs representation. (**b**) Average. (**c**) Classification. (**d**) Correct and wrong classification.

The method proposed above is applied for the coefficients proposed. The following section presents the results obtained.

## 4. Results

### 4.1. Choice of Parameters by Correlation

The coefficients proposed in the previous section have all been calculated on the database detailed in Section 3.2. As the coefficients are numerous, and some are highly correlated with each other (Table 2), only the coefficients that are most representative of the literature, and independent of the others, will be retained for further analysis. It is important to limit the number of parameters to be examined in order to obtain greater clarity, and also to obtain good optimization during PCA.

**Table 2.** Correlation coefficient between each parameter.

|  | $\beta$ | $N_{tot}$ | LWC | $D_{mean}$ | $D_{quad}$ | $D_{vol}$ | $D_{eff}$ |
|---|---|---|---|---|---|---|---|
| $\beta$ | 1.00 | | | | | | |
| $N_{tot}$ | 0.75 | 1.00 | | | | | |
| LWC | 0.98 | 0.63 | 1.00 | | | | |
| $D_{mean}$ | 0.61 | 0.18 | 0.64 | 1.00 | | | |
| $D_{quad}$ | 0.66 | 0.24 | 0.70 | 0.99 | 1.00 | | |
| $D_{vol}$ | 0.69 | 0.31 | 0.73 | 0.96 | 0.99 | 1.00 | |
| $D_{eff}$ | 0.72 | 0.42 | 0.75 | 0.85 | 0.91 | 0.96 | 1.00 |

As shown in Table 2, $D_{mean}$, $D_{quad}$, $D_{vol}$ and $D_{eff}$ are highly correlated. In the following only $D_{mean}$ is retained because it is the most present in the literature. We also find that *LWC* is highly correlated with $\beta$. This is consistent with previous results from the literature [1]. The two parameters *LWC* and $\beta$ are, however, retained because many works in the literature focus on this combination to describe fog. $\beta$ and *V* are directly related to each other. In the following, we preferred to keep $\beta$ because its relation with the other parameters is closer to a linear relation than for *V* (see Equation (3)). This has the advantage of giving better results in PCA. In the continuation, this parameter selection therefore gives: $\beta$, $N_{tot}$, $D_{mean}$ and *LWC*.

Before presenting the results obtained, it is interesting to make a bibliographical review of these different parameters. The $\beta$ and *LWC* coefficients are clearly the most represented in the literature. Figure 6 shows the data available in the state of the art. The points correspond to natural DSD data, for which the authors have given the values of the parameters mentioned. To these natural DSD point data, the most common models in the literature by Deirmendjian [40] and Shettle [41], are added. The data collected as part of the Paris Fog project are also added to Figure 6.

Figure 6 shows that there is a wide dispersion of points. This may be due first of all to real variations in natural fogs. It has indeed been established in the literature that fogs can have different particle sizes, depending on the conditions under which they occur.

However, this dispersion is particularly great for the $D_{mean}$ parameter. This can be explained by the fact that the oldest PSAs were not able to distinguish the smallest drops (<1 µm). These drops, however, represent a very large quantity in the fogs which greatly skews the average. This limitation of the oldest PSA leads to the same problem on the $N_{tot}$ parameter, the latter being underestimated.

Concerning the models in the literature, Deirmendjian's DSDs have very high total concentrations compared to natural data. This can be explained by the fact that when designing these models, they were optimized on truncated DSD data for small drops (again due to the limitations of the first PSA). A large number of small drops was neglected when optimizing the model. When resetting the model in the MOR, the number of small diameter drops therefore exploded, because it was initially not included.

As for the Paris Fog DSDs, these are fairly well-positioned in relation to the data in the literature.

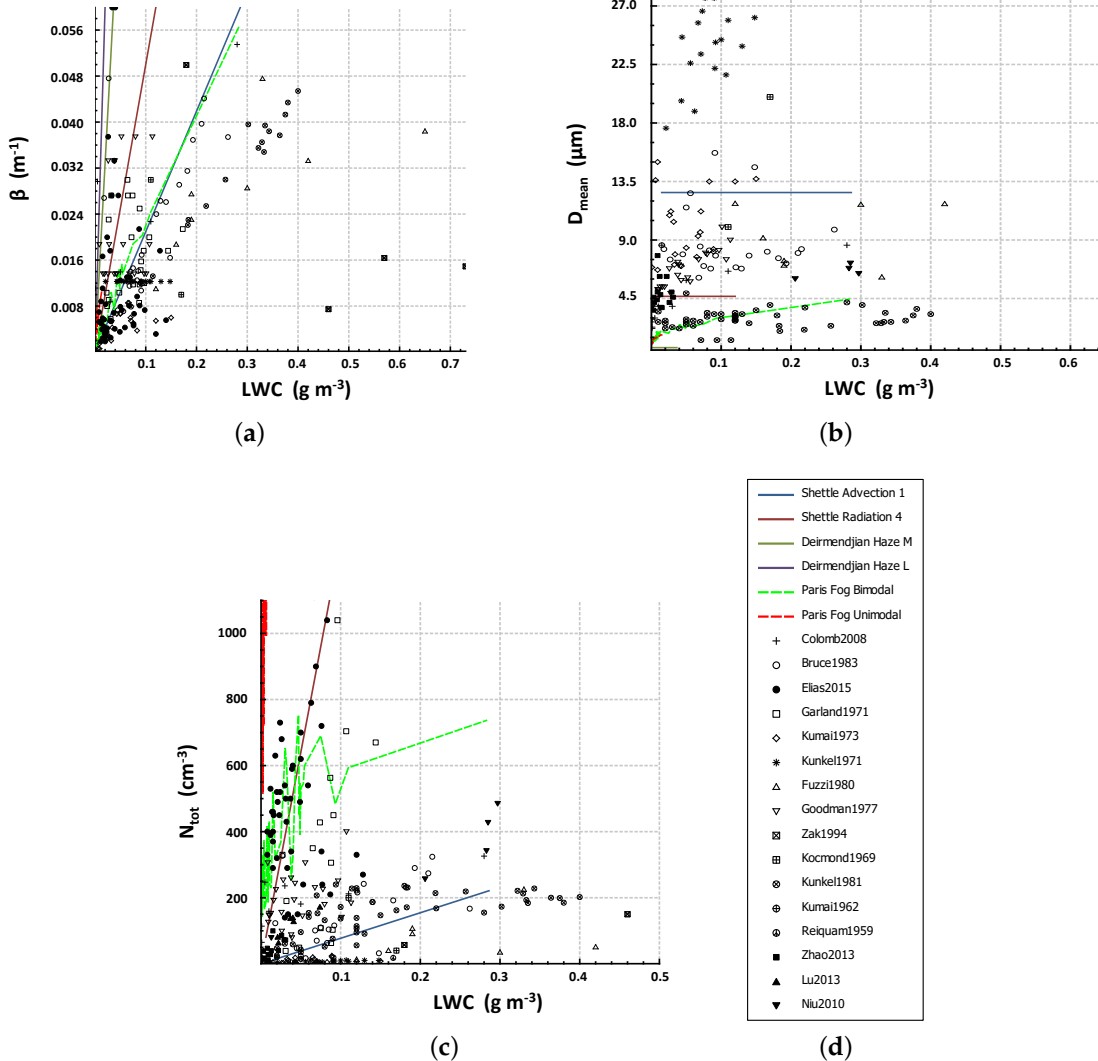

**Figure 6.** Relation between fog droplets parameters as given from the literature (symbols), calculated from Paris Fog data (dashed lines), or from models (continous line). (**a**) *β* vs. *LWC*. (**b**) $D_{mean}$ vs. *LWC*. (**c**) $N_{tot}$ vs. liquid water content (*LWC*). (**d**) Legend.

This review of the literature provides a basis for comparison of natural and modelled fogs. These comparison points can then be used to validate the fogs produced within the platform. This is the subject of the following sections.

## 4.2. Descriptive Analysis

All of the DSDs in the database collected have been processed. Four coefficients were calculated for each DSD and are shown in Figure 7. Figure 7 represents one parameter as a function of another, for all DSDs in the database. In this figure, colored crosses (resp. triangle) represent the data for the MD fog (resp. SD) produced within the platform. A colour scale was added according to the MOR measured by the reference sensor of the platform (completely independent of the PSA). The data represents fog DSDs with MOR values between 10 and 1000 m. In Figure 7, data from the literature (presented in the previous section) was also added, as a counterbalance. In this representation, the literature data were placed into three groups: radiation (black triangle) or advection (black crosses), where the papers mentioned the type of fog measured, and undefined in the case where the source article does not specify which type of fog it is (black circle). Figure 7b,e,g show the zoom of Figure 7a,d,f.

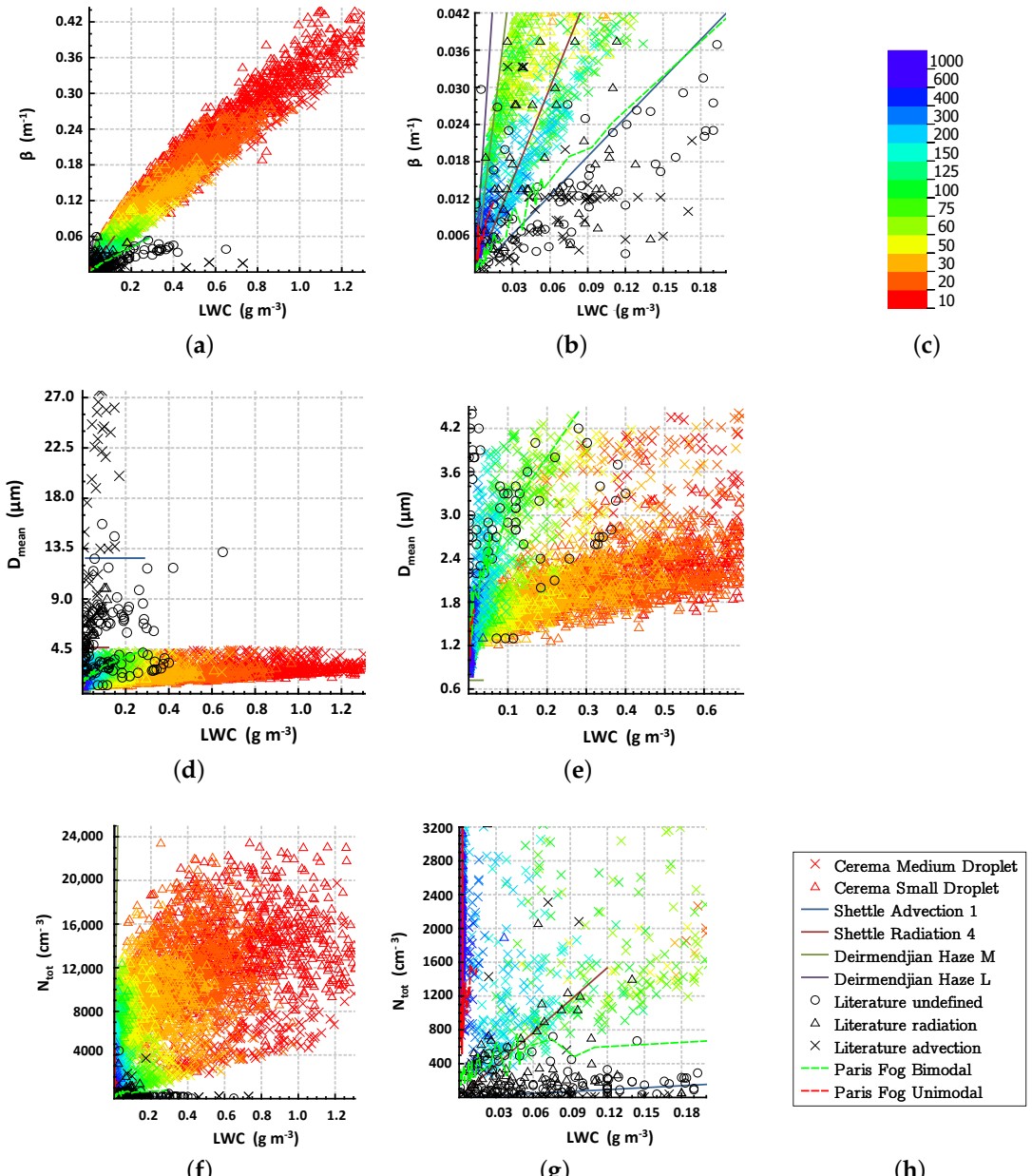

**Figure 7.** Cerema data and state of the art parameters. (**a**) $\beta$ vs. *LWC*. (**b**) $\beta$ vs. *LWC* zoom. (**c**) MOR color bar (m). (**d**) $D_{mean}$ vs. *LWC*. (**e**) $D_{mean}$ vs. *LWC* zoom. (**f**) $N_{tot}$ vs. *LWC*. (**g**) $N_{tot}$ vs. *LWC* zoom. (**h**) Legend.

Figure 7 provides an initial descriptive analysis of the data. We refer to Figure 9 for the quantitative FCC values. Concerning the qualitative classification, as it is shown in Figure 7, first of all the two types of fog produced in the platform (crosses vs. triangles) are visibly different, especially for the coefficients $D_{mean}$ and $N_{tot}$ (Figure 7e,f). Secondly, the *LWC* and $\beta$ coefficients are logically highly correlated with each other (Figure 7a), but also with the reference MOR value (see the colour gradient). The two types of fogs appear to be more different for high MORs (>500 m) than for low MORs. For example, in Figure 7e, the mean diameters are very different for light fog (high MOR, low $\beta$) and gradually come closer together for dense fog (low MOR, high $\beta$): the two groups of dark blue symbols at the lower left of Figure 7e are clearly distinct (one group with $D_{mean}$ in [0.6 μm ; 1.4 μm] and the second group in [1.4 μm ; 2.0 μm]), while the orange and red symbols are close together (some crosses are mixed with triangles). This can be explained by the fact that the less dense fogs are only obtained by dissipation

of initially denser fog. The regulated production of fog (by nozzles) is possible for MORs up to 200 m maximum. The production must then be totally stopped to obtain fogs for MORs greater than 200 m. The phenomenon of natural dissipation and sedimentation then visibly tends to reduce the largest drops first, as the average diameter decreases as the MOR increases. However, this variation seems much slower for the MD fog than for the SD fog on the platform. This is confirmed by Figure 7f where it can be seen that the total number of droplets ($N_{tot}$) varies much more slowly for MD fogs. The two types of fog produced within the platform, therefore, become more and more similar as the MOR decreases, especially under 200 m. However, this is very consistent with natural fogs, where the same trend is observed (Figure 6b).

The overall shape of the point clouds obtained in Figure 7d,f corresponds to what can be found in similar figures in the literature [33–35]. It can also be seen that some fog in the bibliography (natural or model) is well-positioned in relation to the fog produced within the platform. It should be noted that the fogs in the literature are covering a very high MOR range (between 100 and 1000 m) compared to the range of fogs measured within the platform (between 10 and 1000 m). There is therefore no data from the literature for the MOR range between 10 m and 100 m (color of the color-bar between green and red). For example LWC can rise above 0.4 g m$^{-3}$, which is not common in literature. However, in the literature, the natural fogs measured are not as dense as the platform fogs. For this reason, Figure 7b, e.g., zoom in on the areas for which natural fog data from the literature exists. This means that the fogs produced in the platform with visibility between 10 and 100 m cannot be compared to natural fogs, since there are no such data in the literature to our knowledge. Considering the purpose of the platform, that is the evaluation of the performances of vehicle sensors in the most severe conditions, and due to its limited length of 30 m, it is even useful to produce these very dense fog conditions (MOR between 10 and 100 m). Moreover, natural advection fogs have a larger diameter than natural radiation fogs which is perfectly normal, the various references in the literature are therefore consistent. On the other hand, it can be seen that the $D_{mean}$ of the fogs in the literature is much larger than that of the platform. This can always be explained by the limitation of the oldest PSAs but also by the limitation of our PSA which is limited to 17 µm. In addition, the concentrations ($N_{tot}$) obtained in the platform are very high compared to those of fogs in the literature. Indeed, it is rare to see fogs with concentrations above 1000 cm$^{-3}$. This is more consistent with data from clouds, and is also related to the fact that the droplets produced in the platform are very small, which would correspond more to radiation fogs, or even haze-like fogs from areas with polluted air. This results is consistent with LWC values obtained, because $LWC > 0.4$ g m$^{-3}$ is typical for clouds. It suggests that we should work to generate larger droplet sizes in the fogs produced within the platform.

Concerning the models, fog is simulated for MORs between 1000 m and 50 m. Shettle models behave fairly well, with radiation fog having a lower $D_{mean}$ than advection fog. Both types of fog integrate well with the fogs in the literature. It can be seen that platform fog would assimilate very well with the two models proposed by Shettle for the *LWC*, $N_{tot}$ and $\beta$ parameters. The only limitation is that the $D_{mean}$ of the fog produced in the platform is smaller than for that of Shettle. In Figure 7, the Deirmendjian Haze models are not visible because they contain far too many drops, which did not seem valid. The reason for this overestimation was explained in the previous section.

Paris Fog data are interesting in that they are natural data, but also data acquired with the same PSA as those of the platform (PALAS WELAS 2100). These data integrate perfectly with the platform's data from a global point of view. However, it can be seen that they still have DSDs with a $D_{mean}$ a little larger than that of the platform. Following this observation, we will have to work on the development of the platform in order to be able to produce larger droplets.

As a conclusion to this descriptive analysis, the platform proposes fogs that are quite representative of natural fogs and the literature. On the other hand, the proposed DSD range could be extended, especially with fogs with a higher $D_{mean}$. Technical solutions are available and are already being considered to meet this new objective.

### 4.3. Statistical Analysis

The descriptive analysis made it possible to position artificial fogs in relation to natural fogs. However, it is complicated to really classify the two types of fogs, as they are more or less classified according to the parameters analyzed. For example, the parameter pair $D_{mean}$ vs. *LWC* seems to allow a better classification of fog types than the pairs $N_{tot}$ vs. *LWC* and $\beta$ vs. *LWC*. Furthermore, it is difficult to measure how good the classification is. Ultimately, the underlying objective of this study was to clarify which parameters allow better characterization of DSD. As the parameters used are varied, and the literature never uses the same ones, the large database we have at our disposal can help clarify this. In order to answer these problems, a PCA was performed on all the parameters. This allows us to find statistically the best combination of parameters to disperse the available data. In addition, the PCA provides details of the parameter combinations, which makes it possible to know the best parameters for classification of the DSDs. The PCA will, therefore, make reading simpler.

Table 3 presents the results obtained by the PCA. The first PCA vector is a combination of the four parameters, starting with $\beta$. As shown in Figure 8, the first vector is directly related to the MOR. The second vector is rather related to $D_{mean}$ to which $N_{tot}$ is opposed, with very little impact from the other parameters ($\beta$ and *LWC*). The latter is therefore directly related to the type of fog produced. Indeed for a given MOR, the smaller the drops are, the greater their total number, hence the opposition between $D_{mean}$ and $N_{tot}$. It is therefore the second vector that best differentiates the two types of fogs produced in the platform for a given MOR.

**Table 3.** Principal component analysis (PCA) results on conventional parameters.

| PCA Vector Number | Beta | Ntot | LWC | Dmoy | Eigen Value Ratio |
|:---:|:---:|:---:|:---:|:---:|:---:|
| 0 | 0.572028 | 0.485398 | 0.530654 | 0.394436 | 1.0000 |
| 1 | −0.035516 | −0.66099 | 0.0897689 | 0.744159 | 0.2986 |
| 2 | −0.310566 | 0.545225 | −0.563421 | 0.537433 | 0.0718 |
| 3 | −0.758335 | 0.173813 | 0.626821 | 0.0425807 | 0.0025 |

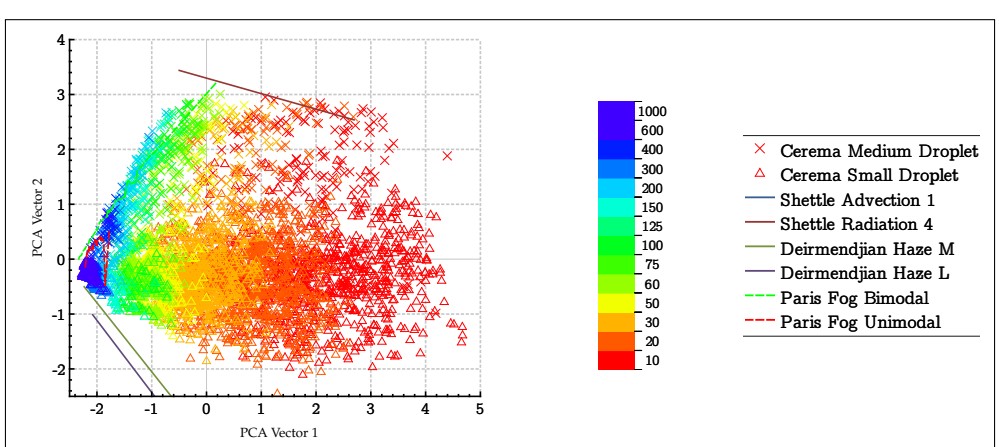

**Figure 8.** Result of the PCA on conventional parameters for various MOR ranges and different datasets.

Figure 8 shows the projected data set for the first two PCA vectors. The dispersion of the data is then visually much clearer. In Figure 8, the literature models and the Paris Fog data are also projected onto the two PCA vectors. The models are then found at the edge of the data from the platform. On the other hand, the fogs collected in the Paris Fog campaign are perfectly included in the point cloud linked to the platform. This is a major result which shows that the platform fogs are well representative of natural fogs. Although data dispersion is better, a metric should be used to compare the different pairs of parameters proposed quantitatively. The method proposed in Section 3.2 was therefore applied to the database. The method is applied in two cases:

- Overall, on all the data.
- By sorting the data by MOR packet (this presupposes having the external reference MOR data, given here by the transmissometer).

Table 4 shows the FCC for each of the two parameters. The pair formed by the first two vectors from the PCA is the one that makes it possible to better classify the data globally, with a differentiation score of 0.93. The $N_{tot}$ vs. $D_{mean}$ pair is, however, quite close to the PCA result, with a score of 0.92. The other parameters are much less relevant for classifying fogs. This result is very interesting and proposes a new vision, since in the literature the parameters most used to distinguish fogs are often $\beta$ and $LWC$.

**Table 4.** Global fog correlation coefficient.

|              |              | FCC   |
| ------------ | ------------ | ----- |
| PCA Vector 1 | PCA Vector 2 | 0.930 |
| $N_{tot}$    | $D_{mean}$   | 0.922 |
| $\beta$      | $D_{mean}$   | 0.831 |
| LWC          | $D_{mean}$   | 0.793 |
| $N_{tot}$    | LWC          | 0.741 |
| $\beta$      | $N_{tot}$    | 0.709 |
| $\beta$      | LWC          | 0.523 |

The overall score allows the parameters to be compared with each other. However, the first descriptive analysis has shown that the DSD coefficients move away from each other when the MOR increases, although this does not mean that the fog would not be differentiated. To verify this, it is possible to go further by applying the method of calculating the FCC on sub-groups of data by classifying them into MOR groups (the reference MOR given by the transmissometer). Figure 9 shows the FCC as a function of MOR. From the figure, it is clear that the $LWC$ vs. $\beta$ pair is not a good combination for classifying fogs by particle size. The other combinations of parameters all obtain FCCs higher than 0.90 for fogs with a MOR higher than 40 m. However, FCCs are worse for fogs of less than 30 m. In the same way, it is generally observed for all parameters that the densest fogs are also the most difficult to differentiate. The best pairs for classifying the fogs for all MORs are the first two PCA vectors and the $N_{tot}$ vs. $D_{mean}$ pair. By spreading out the densest fogs, with MORs below 40 m, the best couples to differentiate the fogs are $\beta$ vs. $N_{tot}$ and $LWC$ vs. $N_{tot}$. Finally, the $N_{tot}$ parameter seems to be the most decisive in the differentiation of fogs, at a fixed MOR. This can be justified, because at a fixed MOR, $N_{tot}$ depends directly on the size of the droplets, with a third-order factor in relation to the diameter. As the MOR is related to the product of the volume of the drops by their number, $N_{tot}$ is highly dependent on the particle size at a fixed MOR. In the case of an analysis of the DSD given the MOR (by a sensor different from the PSA), the PCA did not improve the differentiation of the fog compared to the best combination of coefficients. This analysis grouped by MORs must be supplemented by a global analysis, without the contribution of MORs from an external sensor.

This section has therefore shown that PCA can be of interest in the classification of fogs. In addition, it has shown that the parameter pair $N_{tot}$ vs. $D_{mean}$ is the most relevant for classifying fogs of the platform through the use of FCC.

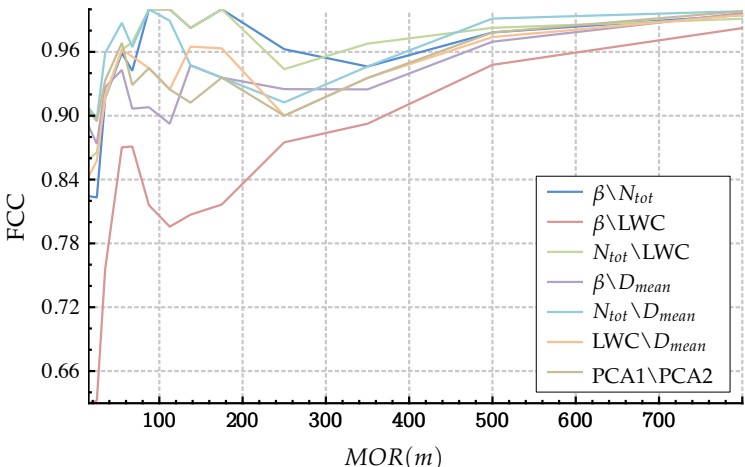

**Figure 9.** Fog correlation coefficient according to the MOR value.

This section has therefore shown that PCA can be of interest in the classification of fogs. In addition, it has shown that the parameter pair $N_{tot}$ vs. $D_{mean}$ is the most relevant for classifying fogs of the platform through the use of FCC.

## 5. Conclusions

The first objective of this study was to show that the platform has two types of fogs with clearly distinct DSDs. To achieve this, we proposed an innovative method, based on the use of a PCA and the FCC. This method is not restricted to a descriptive analysis but also offers a quantitative analysis. It was used on the conventional coefficients characterising DSD in the literature, namely $\beta$, $N_{tot}$, $LWC$, $D_{mean}$, $D_{quad}$, $D_{vol}$ and $D_{eff}$.

In order to compare the platform fogs to natural fogs, several data sources were proposed. A detailed literature review identified macroscopic data (coefficients) from several hundred DSDs of natural fogs, with data acquisition by different PSAs. In addition, the use of data from the Paris Fog campaign made it possible to compare the platform fogs with natural fogs, whose DSD was measured with the same PSA. The use of the most common fog models in the literature provided a third source of comparison. Given the PSA used and the natural data available to us, this study focused on radiation fogs only.

Many results have been achieved. First of all, the two kinds of fog on the platform are clearly different. Using the most relevant coefficients, the FCC obtained is 0.92 overall, rising above 0.95 taking into account the MOR, for MOR fogs above 40 m. In practical terms, this means that we are able to identify which of the two types of fog is reproduced within the platform in 95% of the cases, using only the DSD and the MOR, without any other data. This result, therefore, means that we can confirm that a quantified difference exists between the two types of fog proposed within the platform. Moreover, it appears that the greater the MOR, the more the fogs are identifiable, which may provide development leads to change the DSD of the fogs proposed.

The second result concerns the comparison of the fogs reproduced within the platform with natural fogs: the analysis carried out shows that both types of fogs produced within the platform are included in the range of natural radiation fogs. The fogs proposed are therefore representative of fogs that may exist in nature, both from a macroscopic point of view in terms of MOR, but above all from a microscopic point of view in terms of DSD. In particular, Adverse Weather platform fogs with the smallest drops are at the limit of natural radiation fog or haze-like fog. However, there are examples in the literature of fogs containing much larger droplets and both fogs produced within the platform have quite small droplets, more similar to wet aerosol, and high concentration (closer to polluted fogs or even clouds). Putting aside the fact that older measurements may be skewed due to the use of PSAs that could not measure the smallest droplets and that our sensor cannot measure the largest droplet,

it would, therefore, be worthwhile to develop the platform to find means of producing new fogs with larger droplets. This would suggest extending the range of fogs produced within the platform.

At the same time, this study made possible a quantified analysis of the most representative coefficients in the literature according to their ability to differentiate fogs by their DSD. Since PSAs all have very different characteristics, it is very difficult to compare DSDs from two different devices. By using global coefficients, such as $\beta$, $D_{mean}$, $N_{tot}$, etc., it becomes possible to better compare the fogs according to their DSD. Our study shows that the $N_{tot}$ vs. $D_{mean}$ coefficient pair is the best one to classify the fogs produced in the platform. This result is still consistent with the literature; indeed the most highlighted coefficients are the *LWC* vs. $\beta$ pair, often highlighted for its strong correlation. It would now be very relevant to test this method on a natural fog database.

To conclude, the platform allows systems and sensors to be tested in a large volume of fog (30 m long $\times$ 5 m wide $\times$ 2.2 m height) in fog and rain conditions. The advantage over competitive facilities of similar dimensions is that it benefits from weather conditions that are finely measured and comparable to certain natural conditions as shown in this study (two clearly different fogs that are representative of a part of natural fogs). On the other hand, The Adverse Weather platform is smaller than others proving grounds that allow vehicle tests in high-speed dynamics. Compared to chambers dedicated to the most accurate reproduction of fog conditions (taking into account thermodynamic phenomena and aerosol composition), the platform has the advantage of being much larger, but also of obtaining faster fog variations on demand. On the other hand, it appears in this study that some ranges of fog are missing, in particular the fogs containing larger droplets (diameter > 10 µm), not addressed here.

Then, work will be carried out in order to produce new fogs with DSDs containing larger droplets. The method presented here is entirely reusable for any type of DSD but also for any type of coefficient estimated from the DSD. It can therefore be reapplied: (1) to a set of DSDs from various natural and artificial fogs so as not to be limited to the artificial fogs presented here. In particular, the FCC method would make it possible to check whether fogs can really be differentiated according to the place of occurrence (maritime or continental), the pollution conditions, the climate and the geographical region, or even other artificial or numerical fog simulation platforms; (2) to different parameters, such as extinction coefficients at different wavelengths, or parameters obtained by model optimization on DSDs; (3) to a set of DSDs measured from various PSAs on the same fog. This will be the subject of future work.

**Author Contributions:** Conceptualization, P.D., M.C., and F.B.; data curation, P.D.; formal analysis, P.D.; funding acquisition, M.C.; investigation, P.D.; methodology, P.D.; project administration, P.D., M.C.; software, P.D. and F.B.; supervision, F.B.; validation, P.D., M.C., and F.B.; visualization, P.D.; writing—original draft, P.D.; writing—review and editing, P.D., F.B. and M.C. All authors have read and agreed to the published version of the manuscript.

**Funding:** The research leading to these results has received funding from the European Union under the H2020 ECSEL Programme as part of the DENSE project, Contract Number 692449. This work has been sponsored by the French government research program "Investissements d'Avenir" through the IMobS3 Laboratory of Excellence (ANR-10-LABX-16-01) and the RobotEx Equipment of Excellence (ANR-10-EQPX-44), by the European Union through the Regional Competitiveness and Employment programme, 2014–2020 (ERDF—AURA region), and by the AURA région.

**Acknowledgments:** The authors would like to thank Jean-Luc Bicard for his active participation in this publication.

**Conflicts of Interest:** The authors declare no conflict of interest. The funders had no role in the design of the study; in the collection, analyses, or interpretation of data; in the writing of the manuscript; nor in the decision to publish the results.

## Abbreviations

The following abbreviations are used in this manuscript:

| | |
|---|---|
| ADAS | Advance driving assistance systems |
| DSD | Droplet size distribution |
| FCC | Fog Classification Coefficient |
| LiDAR | Light Detection And Ranging |
| LWC | Liquid water content |
| MD | Medium Droplets |
| MOR | Meteorological optical range |
| PCA | Principal Component Analysis |
| PSA | Particle Size Analyser |
| SD | Small Droplets |
| ToF | Time of Flight |

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
