# Peer review of "Fog Classification by Their Droplet Size Distributions: Application to the Characterization of Cerema’s Platform"

_atmosphere, doi:10.3390/atmos11060596_

Round 1

Reviewer 1 Report

please see attached comments.

Reviewer 3 Report

Summary:

Focusing on the characteristics of artificial fogs produced by the PAVIN BP plateform, the authors try to check them by careful statistical analysis. It is a good job. However, I do not feel this is suitable to be published immediately until the authors answer the following issues in more detail and the caveats of the method are explored.

Comments:

1) the title

Maybe it is better like the following:

Fog classification by their droplet size distributions: application to the characterization of Cerema’s PAVIN BP plateform

2) the abstract

Please add the word artificial/artificially in a suitable place, in order to let readers clearly get the background of the two types of fogs.

3) relationship between fog and transportation
    Fog is one of  major challenges for transportation systems.

4) the first appearance of Cerema

    Suggest use a footnote to introduce Cerema briefly.

5) Lines 41-43

    Maybe here needs some explanations or reasons to show why it is necessary to validate the DSD. I think the formation mechanisms of natural fogs and artificial fogs are different, condensation nucleus are needed in the forming process of natural fogs. 

6) Figure 1

a)  it is very difficult to easily understand the meaning in the Legends. Please make a table to give brief descriptions for them, for example, what the Unimodal means, and so on.

b) Which represent the natural fog observations, besides Paris Fog?

c) Please show more numbers for the axis of droplet radius.

7) Figure 5

There are some obvious errors for the order numbers of panels.

8) Suggestions

a) Please get/show more fog observations about DSD data in natural state.

b) Maybe the method to produce artificial fogs needs to be improved based on the physical formation mechanism of natural fogs, for example, land radiation cooling fog with small DSD and ocean advection cooling fog with big DSD.

Reviewer 4 Report

please see attached file. This paper needs lots of work, and I cannot suggest its acceptance for this journal.

Reviewer 5 Report

Review of the manuscript

Fog classification by their droplet size distributions. Application to the characterization of Cerema’s

PAVIN BP platform.

by Pierre Duthon, Michèle Colomb and Frédéric Bernardin

Marie Mazoyer, Météo-France

The paper titles “Fog classification by their droplet size distributions. Application to the characterization of Cerema’s PAVIN BP platform.” by Duthon et al. aims to characterize types of fog Droplets Size Distribution (DSD) of the Cerema’s platform and show that they are similar to natural fogs. The final goal is to be able to artificially reproduce natural fog’s DSD to then perform a comprehensive study of the fog impact on the car sensors. The method is based on a comparison with DSD issued from literature. Effort is made to determine the more pertinent metrics to describe DSD. In that purpose a principle analysis component technique is used. The determination of pertinent metrics to analyze DSD is of great interest to study fogs microphysic evolution.

Unfortunately, they are an important problem with the literature comparison and so far with your instrumentation.

The authors cite Elias et al. 2015, I refer here to their figure 1. The authors did not consider the largest droplets (blue dots) that the WELAS is not able to measure according to Elias et al. 2015 and also discussed in Mazoyer et al, 2016 (ACP, Experimental study of the aerosol impact on fog microphysics). According to Elias et al. 2015 and Mazoyer et al, 2016, to characterize fog DSD, a CDP or a Fog-Monitor is needed to measure the droplet spectra above ~ 8 µm.

Considering more a “visibility’ issue and so far an extinction issue for your study, as said in Elias et al. 2015: “When fog formed, droplets became the strongest contributors to visible radiation extinction,...”,“The mean transition diameter between the aerosol accumulation mode and the small droplet mode was 4.0±1.1 μm.” and ”consequently, the mean contribution to extinction in fog was 20±15 % from hydrated aerosols smaller than 2.5 μm and 6±7 % from larger aerosols.”

Using only a WELAS in your instrumentation, you don’t measure the entire fog DSD and you miss the largest droplets that get the highest contribution to extinction in fog.

Also, you are trying to resume a distribution by few metrics and to compare it with literature, but I think it should be on the same diameter range than the DSD from literature you compare with, so you should specify the instrumentation used by these studies. Moreover, your literature comparison does not include the most recent research on fog microphysic.

Then, although I found very interesting to use a principle analysis component technique to characterize DSD, I have the feeling that your Table 1 is already giving you a solid answer to which metrics used to describe the fog when looking at the lowest correlation coefficient. Then maybe a same kind of study with all moments of distribution may be interesting, or maybe to characterize the best ‘diameter’ descriptor (Mean, median, volumic, effective, surfacic, another one?). Performing a PAC technique on DSD classes diameter may help to answer these questions.

Eventually, although your finality is not model, I could be interesting to add DSD from model literature to your study.

In conclusion, I think that the manuscript cannot be accepted for publication, in view of the major scientific issue reported above.

Round 2

Reviewer 5 Report

Second review of the manuscript

Fog classification by their droplet size distributions. Application to the characterization of Cerema’s

PAVIN BP platform.

by Pierre Duthon, Michèle Colomb and Frédéric Bernardin

Marie Mazoyer, Météo-France

Significants efforts of clarification have been made. The sections on the PCA have been reduced and discussion has been performed on that. More focus is made on the presentation of the platform. Globally the paper contains more discussions especially on the use of the PALAS and on the high values for Ntot and LWC. However I think that they are still major issues.

1) The firsts ones concern again the literature:

- You still don’t cite the more recent researches on fog microphysics. Some of them already had a reflection on the best metrics to use to define fog DSD and on their correlation between each other. I’m thinking to the very interesting studies of Niu, Zhao, Liu.

- While citing and using Paris-Fog data, you should explain that they are more recents studies on SIRTA’s fog which has been performed with a FM-100 (Burnet, Degefie, ...). And then you should explain that your are using the paper from Elias because you are comparing to work performed with an equal instrumentation.

- While comparing with literature on figure 6 and figure 7, you should present their instrumentation and include more recent papers. Apart from your work which is from 2008, the more recent is 1994. They had been a lot a fog microphysic studies since 1994 and 2008. Or you may explain why you choose these papers.

- You may explain why the Shettle and Deimerdjian models are adapted to your study and explain what do they contain.

2) Your LWC is very high. At the ground we won’t except high LWC values for fog, maybe more on altitude, but no more than 0.4 g.m-3. But you could justify that it is not an issue for you study as your concern is more about visibility and extinction if I understood correctly. Or you could select only DSD with LWC < 0.15-0.2 g.m-3. And justify it with literature support. Or you may clearly write that your study is about fogs and clouds.

3) To validate your instrumentation you may compare the extinction as measured by the visibilimeter and as calculated with the Palas and show plots. You did in a way in section 3.1 but maybe it should be done before and maybe you should discuss it more. Also, a discussion on particule size contribution to extinction may be useful.

4) While I think the PCA method is with no doubt super interesting and get a very high potential on the fog microphysic study. I am still not convinced about what bring the PAC method here. You discuss it in the section 4.2 but if you keep it in the paper, you should explain what is the “added value” you get from it. But you could also accentuate it on the visibility. What are the main parameters that control visibility and extinction. And once these main parameters defined, compare them with literature. You could compare with models for visibility (Kunkel, …). Also you could have consider standard deviation or other moments of the distribution. You didn’t explain well why you choose Dmean, Deff, Dmedian, Ntot, LWC, Bext, … I mean they are others parameters that could be relevant and that are used in other studies like standard deviation.

5) Defining the best descriptors is really super interesting for the community as it is a real issue while chosing one. It is why, I think more discussion is needed on the pros and cons of each descriptor with regard to literature and physic. As it is an objectif of your paper, section 3.2 may be before your method section.

6) Concerning the other sources, I was more thinking of taking the parameters directly in the papers.

Consequently I suggest that you perfom some restructuration, and add more discussion to your paper. The section limitation of the study may appear at the end of the paper with discussion on your results. Moreover I didn’t understood well the figure 4 and the explanation you give about it.

Author Response

see attachement
